# IQSEC2 Deficiency Results in Abnormal Social Behaviors Relevant to Autism by Affecting Functions of Neural Circuits in the Medial Prefrontal Cortex

**DOI:** 10.3390/cells10102724

**Published:** 2021-10-12

**Authors:** Anuradha Mehta, Yoshinori Shirai, Emi Kouyama-Suzuki, Mengyun Zhou, Takahiro Yoshizawa, Toru Yanagawa, Takuma Mori, Katsuhiko Tabuchi

**Affiliations:** 1Department of Molecular and Cellular Physiology, Shinshu University School of Medicine, Matsumoto 390-8621, Japan; mehta.radhaanu@gmail.com (A.M.); yoshirai@shinshu-u.ac.jp (Y.S.); emi_suzuki@shinshu-u.ac.jp (E.K.-S.); 20hm124d@shinshu-u.ac.jp (M.Z.); mori@shinshu-u.ac.jp (T.M.); 2Research Center for Advanced Science and Technology, Shinshu University, Matsumoto 390-8621, Japan; tyoshizawa@shinshu-u.ac.jp; 3Department of Oral and Maxillofacial Surgery, Faculty of Medicine, University of Tsukuba, Tsukuba 305-8575, Japan; ytony@md.tsukuba.ac.jp; 4Department of NeuroHealth Innovation, Institute for Biomedical Sciences, Interdisciplinary Cluster for Cutting Edge Research, Shinshu University, Matsumoto 390-8621, Japan

**Keywords:** IQSEC2, medial prefrontal cortex (mPFC), social behavior, autism, synapse

## Abstract

IQSEC2 is a guanine nucleotide exchange factor (GEF) for ADP-ribosylation factor 6 (Arf6), of which protein is exclusively localized to the postsynaptic density of the excitatory synapse. Human genome studies have revealed that the IQSEC2 gene is associated with X-linked neurodevelopmental disorders, such as intellectual disability (ID), epilepsy, and autism. In this study, we examined the behavior and synapse function in IQSEC2 knockout (KO) mice that we generated using CRIPSR/Cas9-mediated genome editing to solve the relevance between IQSEC2 deficiency and the pathophysiology of neurodevelopmental disorders. IQSEC2 KO mice exhibited autistic behaviors, such as overgrooming and social deficits. We identified that up-regulation of c-Fos expression in the medial prefrontal cortex (mPFC) induced by social stimulation was significantly attenuated in IQSEC2 KO mice. Whole cell electrophysiological recording identified that synaptic transmissions mediated by α-amino-3-hydroxy-5-methyl-4-isoxazolepropionic acid receptor (AMPAR), N-methyl-D-aspartate receptor (NMDAR), and γ-aminobutyric acid receptor (GABAR) were significantly decreased in pyramidal neurons in layer 5 of the mPFC in IQSEC2 KO mice. Reexpression of IQSEC2 isoform 1 in the mPFC of IQSEC2 KO mice using adeno-associated virus (AAV) rescued both synaptic and social deficits, suggesting that impaired synaptic function in the mPFC is responsible for social deficits in IQSEC2 KO mice.

## 1. Introduction

IQ motif and Sec7 domain 2 (IQSEC2), also referred to as Brefeldin A-resistant Arf-GEF 1 (BRAG1), constitutes a family of guanine nucleotide exchange factor (GEF) for ADP-ribosylation factors (Arfs) along with IQSEC1 and IQSEC3. Arfs are a small Ras superfamily GTPase consisting of six members (Arf1-Arf6). In the forebrain, IQSECs selectively activate Arf6 by exchanging their GDP to GTP via the Sec7 domain and promote postsynaptic development [1,2,3,4,5,6,7,8,9,10,11]. IQSEC2 mRNA is strongly expressed in the olfactory bulb, entire cerebral cortex, hippocampus, and cerebellum [1], and the protein is exclusively localized at the postsynaptic density (PSD) of excitatory synapses, where it interacts with various PSD proteins, such as PSD-95, IRSp53, apo-calmodulin, CaMKII, SAP-97, and N-methyl-D-aspartate receptors (NMDARs) [1,12,13,14,15,16].

IQSEC2 has been shown to regulate α-amino-3-hydroxy-5-methyl-4-isoxazolepropionic acid receptor (AMPAR) trafficking through two different pathways. In response to the calcium influx from NMDAR, apo-calmodulin attached to the IQ domain of IQSEC2 is converted to calcium/calmodulin enabling IQSEC2 to activate Arf6 by GEF function. The active form of Arf6 further activates JNK and promotes the internalization of GluA1-containing AMPAR [6,8,14]. This mechanism is associated with activity-dependent long-term depression (LTD) [17]. Another one is the GEF function independent pathway. IQSEC2 enhances the surface localization of GluA2-containing AMPAR. A C-terminal PDZ binding sequence is required for this mechanism [17]. IQSEC2 forms a complex with NMDAR through the interaction with PSD-95. GluN2B subunit is predominantly associated with IQSEC2. In contrast, IQSEC1, another IQSEC family member, is involved in the function of GluN2A-containing NMDAR. Considering the age-dependent subunit switch of NMDAR, IQSEC2 may act for NMDAR function in an earlier stage in the synapse development [16].

IQSEC2 has been implicated in X-linked neurodevelopmental disorders, such as intellectual disability (ID), epilepsy, and autism [8]. Clinical studies have revealed that autistic features are observed in half of the patients with IQSEC2 mutations [8]. In vivo studies have been conducted using loss-of-function and gain-of-function mutant mouse models.

IQSEC2 knockout (KO) mice have been generated from two groups by using CRISPR/Cas9 technology. One was generated by removing exon 3 of the IQSEC2 gene using two single-guide RNAs (sgRNAs) targeting introns before and after the exon [18]. Both hemizygote male and heterozygote female KO mice showed spontaneous seizures. They primarily focused on female heterozygote KO mice for their studies. Female heterozygous KO mice exhibited hyperactivity, reduced neuromuscular strength, increased anxiety, decreased fear response, reduced sociability, decreased novel recognition, and reduced spatial learning and memory [18]. They also studied network activity in dissociated cortical neurons from IQSEC2 female heterozygote KO mice using a multielectrode array system. These neurons exhibited hyperactivity with aberrant synchronicity compared to the control neurons [18]. Another KO mouse line was generated by Jackson Laboratory that attained a frameshift mutation within exon 7 [19]. The hemizygote male KO mice showed spontaneous seizures. However, susceptibility to induced seizures, such as the electroconvulsive threshold test at 6 Hz and the pentylenetetrazole chemoconvulsion test, was suppressed in the IQSEC2 KO mice [19]. These mice also showed hyperactivity in the open field test. The anxiety level in these mice was increased in the elevated plus maze test [19]. No social behavioral test was conducted in these mice.

A350V in IQSEC2 is a missense mutation identified in patients with ID, epilepsy, and autism. The mutation is located in the IQ domain and interferes with the binding with apo-calmodulin [20,21]. A350V behaves as a gain-of-function mutation and the mutant IQSEC2 constitutively activates Arf6. This enhances the internalization of the GluA2 subunit of AMPAR, which is different from the case of the activity-dependent pathway that targets GluA1 [14], resulting in the reduction in the basal synaptic transmission [21]. IQSEC2 A350V mutant mice exhibit hyperactivity and impaired learning and memory ability. Atypical forms, not simple reduction, of social behavior were observed in A350V mutant mice. There was no significant difference or only a subtle reduction in the social preference in the A350V mutant mice [21,22]. The interaction with a stranger mouse was more robust than with a familiar mouse in the mutant mice in the social novelty preference test [21]. Deficits in sex preference and emotional state preference were also observed in the mutant mice, which were rescued by administration of positive AMPAR modulator PF-4778574 [23].

While the molecular function of IQSEC2 has gradually been unlabeled by previous studies, the relevance between IQSEC2 deficiency and the pathophysiology of neurodevelopmental disorders has not been addressed. Here, we studied behaviors and the synaptic functions in IQSEC2 hemizygote KO mice that we independently generated using CRISPR/Cas9-mediated genome editing. We found that IQSEC2 KO mice exhibited autistic behavior, including overgrooming and social impairments. In the patch-clump electrophysiology experiment, we found that the AMPAR, NMDAR, and γ-aminobutyric acid receptor (GABAR)-mediated synaptic transmissions were significantly decreased in the medial prefrontal cortex (mPFC) in the IQSEC2 KO mice. Introduction of IQSEC2 isoform 1 in the mPFC of IQSEC2 KO mice using adeno-associated viruses (AAV) restored the reduced AMPAR, NMDAR, and GABAR-mediated synaptic transmission, as well as social behavioral deficits, suggesting that the impaired synaptic function in the mPFC is responsible for the pathophysiology of the autistic symptoms caused by IQSEC2 mutations.

## 2. Materials and Methods

### 2.1. Animals and Housing Conditions

All animals were group-housed maintaining a 12:12 light-dark cycle (lighting from 08:00 to 20:00) with food and water ad libitum. The room temperature was maintained at 23 ± 2 °C. All procedures of animal experiments were reviewed by the Committee for Animal Experiments and were finally approved by the president of Shinshu University. The methods were carried out in accordance with the Regulations for Animal Experimentation of Shinshu University.

### 2.2. Generation of IQSEC2 KO Mouse

We produced single-guide RNA (sgRNA) targeting exon 3 of IQSEC2 gene (GACUAUCAACCGCUGUGCUC; Integrated DNA Technologies [IDT], IA, USA). The sgRNA and Trans-activating crRNA (tracrRNA; IDT) were dissolved in Opti-MEM (Thermo Fisher Scientific, MA, USA) and incubated at 95 ℃ for 5 min. Then Cas9 protein (IDT) was added to form a ribonucleoprotein (RNP) complex. The RNP complex was transferred into fertilized eggs from C57BL/6J mice (Japan SLC, Shizuoka, Japan) with electroporation (NRPA21; Nepa Gene, Chiba, Japan). The final concentrations of sgRNA, tracrRNA, and Cas9 protein were 200 ng/μL each. The poring pulse was set to voltage: 40 V, pulse width: 3 ms, pulse interval: 50 ms, and the number of pulses: 4. The transfer pulse was set to voltage: 5 V, pulse width: 50 ms, pulse interval: 50 ms, and the number of pulses: ±5. After the electroporation, embryos were transferred to the oviducts of pseudo-pregnant mothers (ICR mice; SLC) and kept until natural delivery. The IQSEC2 mutant mice were screened by genomic PCR with a primer pair (AM-18001-IQSEC2 fwd: 5’-GAACCGTGTAGGCAGTGAAGA-3’/AM-18002-IQSEC2 rev: 5’-ACTGTCCCTCCCTGAATACCA-3’) followed by Sanger sequencing with primer (AM-18002-IQSEC2 rev). A mouse harboring 17 bp deletion in exon 3 was selected and bred with a C57BL6/JJcl strain and backcrossed several generations. They were then bred with a 129+Ter/SvJcl strain and maintained on C57BL6/JJcl and 129+Ter/SvJcl hybrid background. Genotyping was performed by genomic PCR using a set of primers; KT20395: 5′- TCCTGGCTCATTTTATCTCCTCC-3′/KT20396: 5′- GGGTTGGCTCCCAGGACTATC-3′. See Appendix A for primer sequences.

### 2.3. Plasmid Preparation

YS002 = pAAV-H1/U6-hSyn-EGFP: Construction of pAAV-H1/U6-hSyn-EGFP (YS002) was described elsewhere [24]. In short, the H1/U6 multi cloning site cassette from lentiviral vector L309 [25] was cloned into the MLuI site located upstream of the hSyn promoter of the pAAV-hSyn-EGFP (Addgene #50465). YS007 = pAAV-H1/shIQSEC2-hSyn-EGFP: shIQSEC2 oligo DNA pair containing IQSEC2 shRNA target sequence (KT19351: 5'-TCGACCCGGAAGCTATCTATCGGGATAATTCAAGAGATTATCCCGATAGATAGCTTCCTTTTTTGGAAAT-3′/KT19352: 5′-CTAGATTTCCAAAAAAGGAAGCTATCTATCGGGATAATCTCTTGAATTATCCCGATAGATAGCTTCCGGG-3′) [16] was annealed and cloned into the XhoI/XbaI site downstream of the H1 promoter of YS002. YS036 = pAAV-EFS-Flag-IQSEC2: Total RNA purified from rat forebrain (Wistar, 6-week-old male) using ISOGEN (NIPPON GENE, Toyama, Japan) was subjected to reverse transcription using SuperScript II Reverse Transcriptase (Invitrogen, Carlsbad, USA) and Oligo (dT) primer to obtain rat brain cDNAs. Flag-tagged IQSEC2 was amplified by PCR from rat brain cDNAs using PrimeSTAR Max DNA polymerase (Takara, Kusatsu, Japan) and a primer pair B (Flag_IQSEC2_F: 5’-ACCGGTGCCACCATGGACTACAAAGACGATGACGA-3’/IQSEC2_R: 5’-TCTTTTATTGAATTCTCAGACCACAGTGCTGA-3’). SpCas was removed by digestion with NcoI and EcoRI from pAAV-EFS-SpCas9 (#104588 Addgene, Watertown, USA). A 4.5bp DNA fragment encoding Flag-IQSEC2 was cloned into the NcoI/EcoRI site of pAAV-EFS-SpCas9 using InFusion HD Cloning Kit (Takara). AM004 = pAAV-EFS-Flag-IQSEC2 (shIQSEC2-Resistant): IQSEC2 shRNA resistant mutation was introduced into YS036 by site-directed mutagenesis with primer pair C (IQSEC2_shR_F: 5’-GCAATTTACCGAGATAAGGAGCGAGAAGCTTCC-3’/IQSEC2_shR_R: 5’-ATCTCGGTAAATTGCTTCCCGGTTCTGGTAAGC-3’) using PrimeSTAR Mutagenesis Kit (Takara, Japan). AM006 = pAAV-EFS (control): Flag-IQSEC2 was removed from AM004 by digesting NcoI and EcoRI, blunt-ended by Klenow fragment (Takara, Japan), and self-ligated. All oligo primer sequences used in this study are listed in Appendix A.

### 2.4. Evaluation of IQSEC2 Knockdown (KD) Efficiency

Mouse cortical neuron cultures were infected with AAV-H1/shIQSEC2-hSyn-EGFP or AAV-H1/U6-hSyn-EGFP at DIV2 and harvested at DIV13 or DIV14. Total RNA was extracted using RNAiso (Takara) following the manufacturer's instructions. 1 μg RNA was reverse transcribed to cDNA using the High Capacity cDNA Reverse Transcription Kit (Applied Biosystems, Waltham, USA). 50 ng of cDNA was subjected to qPCR using Power SYBR Green PCR Mix (Applied Biosystems) on QuantStudio3 (Thermo Fisher) with KT19374: 5′-CGGAGGTCACAGCACCAGTAC-3′ and KT19375: 5′-CCTCCACACTGACTGTTCTG-3′ for IQSEC2 or GAPDHfwd: 5′-CATGGCCTTCCGTGTTCCTA-3′ and GAPDHrev: 5′-CCTGCTTCACCACCTTCTTGA-3′ for GAPDH.

### 2.5. Viral Preparation and Titration

For the production of AAV vectors, we used the triple transfection method. Fifty to seventy percent confluent AAVpro 293T cells grown in DMEM supplemented with 10% FBS were transfected with pHelper plasmid (20 mg/dish), pRC-DJ plasmid (11 mg/dish), and pAAV-IQSEC2 plasmid (10 mg/dish) using acidified PEI in 5 15-cm dishes. Two days after transfection, cells were collected and centrifuged at 1000 rpm for 5 min. The supernatant was removed, and the cells were resuspended in PBS and subjected to a freeze-thaw cycle 3 times. Following the addition of Benzonase (SIGMA-Aldrich), the cell lysate was incubated at 37 °C and centrifuged at 3600 rpm for 15 min. The supernatant was further purified using discontinuous iodixanol gradient ultracentrifugation using Optima XE-90 Ultracentrifuge (BECKMAN COULTER). AAV solutions were titrated by real-time quantitative PCR (Applied Biosystems) using primers KT18345/KT99100 for EGFP, KT19385/KT19363 for the hSyn promoter, or EFS1-F/EFS1-R for EFS promoter. Titers are expressed as viral genomes per mL (vg/mL). The titers for each AAV were 1.13 × 10^11^ (vg/mL)(pAAV-EFS-Flag-IQSEC2), 1.40 × 10^11^ (vg/mL)(pAAV- EFS), and 0.9 × 10^10^ (vg/mL)(pAAV-H1/U6-hSyn-EGFP).

### 2.6. Immuno-Blotting

Brain lysates were prepared from P37 male mouse forebrains (2 WT mice and 2 KO mice, independently) with a RIPA buffer (20mM HEPES pH 7.4, 100mM NaCl, 1mM EDTA, 1% Triton X-100 containing Protease Inhibitor cocktail (Nacalai, Kyoto, Japan)) using a probe sonicator (TOMY UD-201). The protein concentration of the brain lysate was quantified by BioRad Protein Assay System (BioRad, Hercules, USA). Protein in the amount of 0.1mg was subjected to SDS-PAGE (7.5% LaemmLi) and electro-blotted onto Immobilon-FL PVDF membrane (Millipore, Burlington, USA). Western blotting was carried out using anti-IQSEC2 rabbit polyclonal antibody (PAS72831, 1/2000 dilution, Thermo Fisher Scientific,Waltham , USA) and anti-β-Actin mouse monoclonal antibody (MBL, M177-3, 1/5000 dilution), as primary antibodies. IRDye 800CW Goat anti-Rabbit secondary antibody (LI-CDR, 1/5000 dilution) and IRDye 800 Goat anti-Mouse secondary antibody (LI-CDR, 1/5000 dilution) were used as secondary antibodies. Infrared signals were detected by ODYSSEY Imaging System (LI-CDR Bioscience, Lincoln, NE, USA).

### 2.7. In Utero Electroporation

In utero electroporation was performed as described previously [26,27,28]. Briefly, the Institute of Cancer Research (ICR) pregnant mice with E15.5 embryos were anesthetized, and the uterine horns were exposed. Approximately 1 μL of 1 μg/μL pAAV-H1-hSyn-EGFP (Control)/pAAV-H1-shIQSEC2-hSyn-EGFP (IQSEC2-KD) plasmid DNA solution was injected together with 0.01% Fast green for visualizing delivery into the lateral ventricles of the embryos. The embryos were then subjected to a 5-time-repeat of square electric pulses (35 V, 50 ms, 1 Hz) using an electroporator (CUY21E; NEPA Gene, Chiba, Japan) and returned to the uterus for normal maturation. After delivery, positively transfected pups were screened by EGFP signals through the scalp using a blue LED handy light with a filter. On postnatal day 14–19, morphologically normal brains were taken for electrophysiology and histology.

### 2.8. Stereotaxic Surgery

P21-23 IQSEC2 KO or WT mice were anesthetized with an intraperitoneal injection of an anesthesia cocktail (3% Dexmedetomidine Hydrochloride + 8% Midazolam and 10% Butorphanol Tartrate by volume in saline; 0.1 mL/10gm body weight of mouse). Approximately 300 nL of AAV (3:1 by volume of Rescue Virus (AAV-EFS-IQSEC2): AAV-H1-hSyn-EGFP virus/3:1 by volume of Control Virus (AAV-EFS): AAV-H1-hSyn-EGFP virus) was injected bilaterally in the following stereotactic co-ordinate of mPFC- anterioposterior (AP) + 1.75 mm, mediolateral (ML) ± 0.35 mm, and dorsoventral (DV) − 1.75 mm taking bregma as the reference point using a stereotaxic device (Narishige, Tokyo, Japan). Two weeks after AAV injection, the animals were used for experiments.

### 2.9. Electrophysiology

Irrespective of the age of mice, patch-clamp recording from acute brain slices was performed as described previously [29,30]. Briefly, postnatal day 14–19 or postnatal day 35–40 the mice were shortly anesthetized with Isoflurane and decapitated. The brains were rapidly removed and placed immediately in ice-cold slicing artificial cerebrospinal fluid (ACSF) (in mM: 85 NaCl, 75 sucrose, 2.5 KCl, 1.25 NaH_2_PO_4_, 24 NaHCO_3_, 25 glucose, 0.5 CaCl_2_, and 4 MgCl_2_) saturated with 95% O_2_/5% CO_2_ for 2 min. Maintaining the temperature to near 0 °C, the brains were trimmed coronally with razor blades and placed in a vibratome chamber (Campden 7000smz). Three hundred fifty-μm-thick coronal sections constituting the medial prefrontal cortex were transferred to a recovery chamber filled with recording ACSF (in mM: 126 NaCl, 2.5 KCl, 1.25 NaH_2_PO_4_, 26 NaHCO_3_, 10 glucose, 2 CaCl_2_, and 2 MgCl_2_), followed by incubation at 32 °C for 30 min, and then at room temperature for 1 h before recording, maintaining aeration of solution as earlier. Acute brain slices were transferred to a recording chamber continuously superfused with oxygenated ACSF (1.5 mL/min) maintained at 30.5 °C. For patch-clamp electrophysiological recording, pyramidal neurons in layer 5 of the mPFC were identified morphologically using an infrared-differential interference contrast microscope (BX50WI; Olympus, Tokyo, Japan) with a ×40 water immersion objective and a charge-coupled device camera (C3077-79; Hamamatsu Photonics, Hamamatsu, Japan). Postsynaptic responses were measured in voltage-clamp mode using cesium-based ICS (in mM: 130 CsOH, 130 Gluconic acid, 6 CsCl, 10 HEPES, 1 EGTA, 2.5 MgCl_2_, 2 magnesium ATP, 0.5 sodium GTP, 10 phosphocreatine sodium, 290 mOsm). To record mEPSCs (Glutamatergic), cell membrane potential was held at -60 mV for 3-min recording, and then cell membrane potential was shifted to 0 mV to record mIPSCs (GABAergic) from the same cell for 3 min as well. Evoked postsynaptic currents were triggered with 0.1 msec current injections by a nichrome-wire electrode placed at a position around 150 mm from the soma of neurons recorded. For evoked AMPA-EPSC, 100 μM picrotoxin and 50 μM D-AP5 were added in a bath solution. Peak amplitude at −70 mV holding potential was measured for AMPA-EPSC. For GABA-IPSC, 20 μM of DNQX was added in the bath solution with holding potential at −70 mV. For NMDA-EPSC, 100 μM picrotoxin and 20 μM DNQX was added in the bath solution with holding potential at 40 mV. In the recording of evoked synaptic inputs, we used 5 μM of QX-314 in the pipette solution to block sodium channel mediated currents. To calculate the NMDA to AMPA ratio, the amplitude of NMDA current at 50 ms after the onset was divided by the peak amplitude of AMPA current. Paired-pulse ratio (PPR) was recorded by holding the cell at −60 mV for excitatory PPR (ePPR) in the presence of 100 μM picrotoxin. For inhibitory PPR (iPPR), the cells were held at 0 mV in the presence of 20 μM DNQX. All PPR experiments were conducted with (30, 50, and 100 ms) stimulation intervals. Access resistance was monitored throughout the recording, and cells with access resistance over 25 MΩ were rejected. All data were acquired at 10 kHz with EPC10 double amplifier (HEKA) operated by Patch Master software (HEKA). Data analysis was performed with Mini Analysis Program (Synaptosoft, Portland, OR, USA) and custom-made programs of Igor Pro (WaveMetrics, Portland, OR, USA).

### 2.10. Behavioral Test

Behavioral tests were conducted between 10:00 and 19:00. All experiments were performed on IQSEC2 KO and their wild-type littermate mice at 8–16 weeks of age. For AAV-mediated rescue and overexpression experiments, each AAV was injected bilaterally into the mPFC 2 weeks before behavioral tests. Experimenters were blinded to genotype and viral treatments. For all behavioral tests, mice were exposed to a 10 min habituation phase in an empty plastic cage without bedding.

#### 2.10.1. Open Field Test

Each mouse was placed in the corner of the open field apparatus (50 × 50 × 40 cm). The apparatus was surrounded by a sound-attenuating white chest and illuminated at approximately 100 lux. Subject behaviors were recorded from above the apparatus using a CCD camera (WAT-902B; Watec, Yamagata, Japan). Analog images were converted to digital images (720 × 480 pixels) using Monster HD264 (SKNET, Yokohama, Japan). The video frame rate was 30 frames per second (fps). The test lasted 30 min. We measured the travel distance, time spent in the center area (25 × 25 cm), vertical activity (rearing and leaning), and grooming. The travel distance was analyzed using idTracker [31]. Other behavioral parameters were analyzed by a trained observer who was blind to the conditions of the mice.

#### 2.10.2. Elevated Plus-Maze Test

The elevated plus maze consisted of 2 open arms (25 × 5 cm) and 2 closed arms of the same size with 15-cm high transparent plastic walls. The maze was arranged in a manner such that arms of the same type were opposite each other, connected by a central area (5 × 5 cm), and the entire maze was elevated to a height of 50 cm above the floor. In order to keep the mice from falling over, the open arms were surrounded by a 3-mm high edge. The animals were placed individually in the center of the maze, facing a closed arm. Mouse behaviors were recorded during a 5-min test period using a web camera HD Webcam C615 (Logicool, Tokyo, Japan). The video images (640 × 480 pixels) were recorded at 30 fps and analyzed using idTracker. The percent time spent in the open arms relative to the total time spent both in the open and closed arms, and the number of entries into the open and closed arms was analyzed.

#### 2.10.3. Object Recognition Test

The apparatus was a rectangular, 3-chambered box. The chamber was 20 × 40 × 25 cm, and the dividing walls were made from transparent Plexiglas, with small openings (5 × 3 cm) allowing access into each chamber. After the 10-min-habituation period, the mouse was placed in the central chamber and allowed to explore the whole chamber for 10 min (the doorways into the 2 side chambers were opened). Identical sample objects (A1 and A2) were placed in the 2 side chambers. The mice were then allowed to explore the whole chamber for 10 min (familiarization phase). After the familiarization phase, sample object (A3) and novel object (B) were placed on the 2 sides of the chambers, and the mice were again allowed to explore the whole chamber for 5 min (test phase). The location of sample and novel objects in the left or right-side chambers was systematically alternated between trials. The amount of time that the subject head was within a 2-cm distance of the object was measured as “time spent around an object.”

#### 2.10.4. Social Interaction with a Juvenile Mouse

One day before the test, subject and stimulus mice were individually placed into a test arena (50 × 50 × 40 cm), and allowed to explore the arena for 10 min. Stimulus animals were 3–4 week-old male wild-type C57BL/6JJmsSlc mice. On the test day, subject and stimulus mice were placed together into the arena, and their behaviors were recorded for 10 min using a CCD camera. The time spent in social contact by the subjects was analyzed by a trained observer. Behaviors that were scored as social contact included the following: sniffing, grooming, contact with nose, and following. Naïve stimulus mice were used in all encounters.

#### 2.10.5. Sociability and Social Novelty Preference Tests

The method was similar to the object recognition test except for using a stranger mouse instead of an object. We used a 3-chambered box. After the 10-min habituation period (the doorways into the 2 side chambers were opened), the 2 side chambers contained an inverted empty small black wire cup. A clear glass cylinder was placed on top of the inverted cup to prevent lifting or climbing on top. Following the habituation period, the mice were placed back into the central chamber, and the doorways into the 2 side chambers were closed. In the sociability test, an unfamiliar male mouse (stranger 1, S1) with no prior contact with the subject mouse was placed in 1 of the 2 cups, and then the doorways were unblocked. The location of S1 in the left or right-side chambers was systematically alternated between trials. The subject behaviors were recorded for 10 min using a CCD camera. After the sociability test, the subject mouse was again confined in the central chamber. In the social novelty preference test, a second unfamiliar male mouse (stranger 2, S2) was enclosed in the cup that had been empty (E) during the sociability test, and the doorways were again unblocked. The stranger mice were at least 2 weeks younger than the subject mice. In both tests, the amount of time that the subject head was within a 2-cm distance of the wire cup was measured as “time spent around cup”.

#### 2.11. c-Fos Mapping

c-Fos mapping in mouse brain was performed essentially as described previously [32]. IQSEC2 KO and wild-type littermate mice were divided into 2 groups. One group was imposed to social and social novelty preference test in a 3-chamber apparatus (mice with social stimulation), and another was kept quietly in home cages (mice without social stimulation). 90–120 min after completion of the social tests in the mice with social stimulation, both groups of mice were perfused transcardially with PBS (pH 7.4), followed by 4% paraformaldehyde in PBS. Thirty-micrometer-thick coronal sections were prepared with a cryostat CM1950 (Leica). The sections were incubated at room temperature in PBS containing 0.25% Triton-X-100; blocked in 10% of normal donkey serum in PBS containing 0.005% Tween 20 (PBST) at RT for 1 h; and incubated at 4 °C with rabbit anti-c-Fos antibody (ab190289, 1:5000, Abcam, Cambridge, UK) in PBST. After overnight incubation at 4 °C with primary antibodies, the brain sections were washed with PBST and incubated with Alexa 594-conjugated donkey anti-rabbit IgG (Abcam, ab150076, 1:1000), for 2–3 h at room temperature. After further washing with PBS, brain sections were then stained with DAPI for 5 min and mounted on chamber slides with a coverslip. Fluorescence images were taken with an all-in-one fluorescent microscope (BZ-X710, Keyence) and a confocal laser-scanning microscope TCS SP8 (Leica Microsystems). Referring to the Paxinos Brain Atlas and using the GNU Image Manipulation Program (GIMP), c-Fos positive brain regions were identified and located. For quantification, the background was subtracted from confocal images and adjusted for brightness and contrast uniformly to count c-Fos-positive cells using the cell counter function of ImageJ (NIH, Bethesda, USA). An average of 2–4 brain slices per mouse was compared between groups for each brain region analyzed. Three animals were analyzed for each group.

### 2.12. Imaging and Histology

Infection of AAV in the mPFC was confirmed histologically by monitoring GFP signals in brain slices. Transcardial perfusion and fixation of mouse brain were conducted as described in c-Fos mapping section. Fixed mouse brains were coronally sectioned at 100 μm-thickness with a cryostat CM1950. The brain sections were stained with DAPI for 5 min and mounted on chamber slides with a coverslip. GFP signal was examined in the brain sections, including mPFC (Infralimbic Cortex, Prelimbic Cortex, and Anterior Cingulate Cortex) using an all-in-one fluorescent microscope (BZ-X710, Keyence, Osaka, Japan) and a confocal laser-scanning microscope (TCS SP8; Leica Microsystems, Wetzlar, Germany). Mice expressing fluorescent signals in the mPFC were exclusively used for data analysis.

### 2.13. Sample Size and Statistical Analysis

Sample sizes were determined based on established practice and on our previous experience in respective assays. The number of independent samples (e.g., neurons/animals or animals in behavioral experiments) was indicated on or below the graphs. All values represent the average of independent experiments ± SEM. The variance among analyzed samples was similar. Statistical significance was determined by Student’s *t*-test (for 2 groups) or one-way ANOVA followed by Tukey’s post-hoc test in case of behavioral experiments and c-Fos mapping and Bonferroni’s post-hoc test in case of electrophysiology experiments (for multiple groups). Statistical analysis was performed using an online calculator (https://astatsa.com, accessed on 1 July 2021), and heat maps were created using custom-written R scripts. Detailed p values, means, and SEM are listed in Appendix A. Statistical significance was indicated by asterisks (* *p* < 0.05, ** *p* < 0.01, *** *p* < 0.001). All data were expressed as means ± SEM.

## 3. Results

### 3.1. Generation of IQSEC2 KO Mice

To generate IQSEC2 KO mice, we designed an sgRNA within exon 3 of the IQSEC2 gene, a common first exon of isoform 1 and 2 (Appendix A). We electroporated 1-cell stage fertilized eggs of the mouse with the sgRNA and Cas9 mRNA and transplanted the eggs into pseudo-pregnant foster mothers to obtain genome-edited mice. By genomic PCR followed by direct sequencing, we identified a mouse harboring 17 bp deletion within exon 3, introducing a stop codon after the serine at position 254 (Appendix A). We selected this mouse and backcrossed it to the C57BL/6 strain to establish an IQSEC2 KO mouse line. We confirmed that the IQSEC2 protein was completely removed from the mouse brain by Western blotting (Appendix A). Because of the difficulty in the breeding on the C57BL/6 background, we decided to cross these mice to the 129/SvJ strain and maintained them on the C57BL/6 and 129/SvJ hybrid background. We used hemizygote male KO mice and wild-type littermates on the hybrid background for all experiments.

### 3.2. IQSEC2 KO Mice Exhibit Hyperactivity, Overgrooming, and Impaired Social Behavior

To study the relevance between IQSEC2 deficiency and neurodevelopmental disorders, we studied behaviors in IQSEC2 KO mice. In the open field test, travel distance was significantly increased in the IQSEC2 KO mice, suggesting these mice were hyperactive (Figure 1A). The IQSEC2 KO mice showed overgrooming (Figure 1A), a sign for repetitive and compulsive behavior typical for autism. The time spent in the center and the vertical activity that indicate anxiety level were unaltered in the IQSEC2 KO mice (Figure 1A). Consistent with this, times spent in the open and closed arms in the elevated plus maze, a standard behavioral paradigm for anxiety test, were equivalent between wild-type and KO mice (Figure 1B).

We next studied social behavior in IQSEC2 KO mice using a three-chamber apparatus. First, we examined the preference between a stranger mouse (S1) and an empty cage (E1) for measuring social preference. While wild-type mice interacted with a stranger mouse more than an empty cage, the interaction time between a stranger mouse and an empty cage was equivalent in IQSEC2 KO mice (Figure 1C). To measure the social novelty preference, we placed a new stranger mouse (S2) in the empty cage in the same three-chamber apparatus and examined the interaction time in the same groups of test mice. Since the former stranger mouse (S1) was already exposed to the test mice, it was considered to be a familiar mouse in this experiment. Wild-type mice interacted more time with a new stranger mouse than a familiar mouse, but the preference for a new stranger mouse was diminished in the IQSEC2 KO mice (Figure 1D). These experiments suggest that both social preference and social novelty preference were impaired in the IQSEC2 KO mice. We also examined the social interaction with a free-moving juvenile mouse. The interaction time with a juvenile mouse was significantly decreased in the IQSEC2 KO mice (Figure 1E). We examined the preference for a novel object in the three-chamber apparatus and found that the interaction with a novel object was significantly increased in the IQSEC2 KO mice (Figure 1F).

### 3.3. Up-Regulation of c-Fos Expression in the mPFC by Social Stimulation Is Diminished in IQSEC2 KO Mice

To identify the brain region responsible for the social deficit in the IQSEC2 KO mice, we examined c-Fos expression in brain slices from wild-type and IQSEC2 KO mice with or without the social task. In wild-type mice, c-Fos expression was increased in various brain regions, including infralimbic cortex, prelimbic cortex, cingulate area 1, cingulate area 2, lateral septum, central amygdala, posterior paraventricular thalamus, lateral habenula, medial dorsal thalamus, periaqueductal gray, and anterior paraventricular thalamus (Figure 2C). However, the increase was attenuated in the infralimbic cortex, prelimbic cortex, cingulate area 1, and cingulate area 2 in IQSEC2 KO mice (Figure 2A–C). This suggests that activation of neural function in the mPFC is required for exerting proper social behavior, and this mechanism may be impaired in IQSEC2 KO mice.

### 3.4. AMPAR, NMDAR, and GABAR-Mediated Synaptic Transmissions Are Decreased in the mPFC of IQSEC2 KO Mice

IQSEC2 is strongly expressed in the entire cerebral cortex, including mPFC [1]. Based on the result of c-Fos mapping before and after the social task, we hypothesized that the neural function in the mPFC was involved in the social deficits in the IQSEC2 KO mice. To address this hypothesis, we examined the synaptic function in pyramidal neurons in layer 5 of the mPFC, of which neural circuits have been suggested to be involved in social behavior [33,34,35], in acute slices prepared from young (P14–P19) wild-type and IQSEC2 KO mice. We first measured miniature synaptic currents in the presence of tetrodotoxin. In this recording, we found the frequency, but not amplitude, of both mEPSCs and mIPSCs were significantly decreased in IQSEC2 KO mice (Figure 3A,B). Decrease of mEPSCs/mIPSCs frequency often reflects the reduction of release probability of synaptic vesicles. To verify this possibility, we examined the PPR of evoked synaptic transmission in wild-type and IQSEC2 KO neurons. We found the ePPR was significantly increased in the IQSEC2 KO neurons (Figure 3C), suggesting decreased release probability may contribute to the mechanism for the reduction of mEPSC frequency. We failed to detect changes in iPPR (Figure 3D). Next, we examined the synaptic strength by measuring the amplitude of evoked synaptic transmission. We found that AMPAR, NMDAR, and GABAR-mediated evoked synaptic transmissions were decreased in IQSEC2 KO mice (Figure 3E–G). To examine the degree of impairment between AMPAR and NMDAR functions, we measured the NMDA/AMPA ratio. In this experiment, we found that the NDMA/AMPA ratio was significantly increased in IQSEC2 KO mice (Figure 3H), suggesting that the impairment in AMPAR was more robust than that in NMDAR. We also studied all these parameters in adult (P35) IQSEC2 KO mice and obtained the same results (Appendix A).

### 3.5. Postsynapse Specific IQSEC2 KD Reproduces Decreased AMPAR, NMDAR, and GABAR-Mediated Synaptic Transmissions

IQSEC2 has been shown to be an exclusive postsynaptic protein. Decreased release probability raised whether this was due to a presynaptic function of IQSEC2 or postsynaptic IQSEC2 affecting the opposed pre-synapse development. To address this question, we knocked down IQSEC2 expression specifically in postsynaptic neurons by in utero electroporation with IQSEC2 shRNA. We adapted an shRNA sequence from the previous paper that targeted within exon 1 of IQSEC2 gene [16] and cloned it into a plasmid vector downstream of H1 promoter. EGFP cassette driven by human synapsin 1 promoter (hSyn) was also cloned in the same vector for transfection marker. The vector lacking the shRNA was used as a control (Figure 4A). Due to technical accessibility, we introduced shRNAs sparsely into pyramidal neurons in layer 2/3 of the somatosensory cortex (Figure 4B).

As we observed in the IQSEC2 KO mice, the frequency of mEPSCs and mIPSCs was significantly decreased in IQSEC2 KD neurons (Figure 4C,D). Unlike IQSEC2 KO mice, the amplitude of mEPSCs, but not mIPSCs, was also decreased in the IQSEC2 KD neurons (Figure 4C,D). We examined ePPR and found the release probability was decreased in the IQSEC2 KD neurons (Figure 4E). We also examined the evoked synaptic transmission and found that AMPAR, NMDAR, and GABAR-mediated synaptic transmissions were significantly decreased in IQSEC2 KD neurons, as observed in IQSC2 KO mice (Figure 4F–H). The NMDA/AMPA ratio was also increased in the IQSEC2 KD neurons (Figure 4I). These results suggest that the synaptic phenotype observed in the IQSEC2 KO mice is due to the postsynaptic effect of IQSEC2 deficiency.

### 3.6. Reexpression of IQSEC2 in the mPFC Rescues the Impaired Synaptic Transmission and Social Behavior in IQSEC2 KO Mice

The next question is whether the impaired synaptic function in the mPFC is responsible for the behavioral deficits. To address this question, we introduced the full-length IQSEC2 isoform 1 gene specifically in the mPFC using AAV (Figure 5A). First, we examined whether the re-expression of IQSEC2 recovers the synaptic functions. In this experiment, we found the decrease of the frequency of mEPSCs and mIPSCs was restored to the wild-type level in the IQSEC2-AAV infected neurons (Figure 5B,C). The decreased amplitudes in evoked AMPAR, NMDAR, and GABAR-mediated synaptic transmission were also recovered in the IQSEC2-AAV infected neurons (Figure 5D–F). The paired-pulse ratio was also recovered in IQSEC2-AAV infected neurons (Figure 5G).

We then injected IQSEC2-AAV in both hemispheres of the mPFC in IQSEC2 KO mice and examined the social behaviors. The interaction with a free-moving juvenile mouse was recovered in IQSEC2-AAV infected mice (Figure 6A). IQSEC2-AAV infected mice interacted significantly more time with a stranger mouse (S1) than empty cage (E) in the social preference test (Figure 1B). Interaction with a new stranger mouse (S2) became higher than familiar mouse (S1) in IQSEC2-AAV infected mice (Figure 6C), suggesting sociability and social novelty preference were also restored. We examined whether simple overexpression of IQSEC2 isoform 1 in the mPFC in wild-type mice affects the behaviors or not (Appendix A). We found that the overexpression of IQSEC2 in the mPFC unaltered the anxiety (Appendix A), social interaction (Appendix A), social preference (Appendix A), social novelty preference (Appendix A), and novel object preference (Appendix A), suggesting that overexpression of IQSEC2 does not affect social behaviors.

## 4. Discussion

In this study, we found that 1. IQSEC2 KO mice exhibited autistic behaviors, including overgrooming, decreased social interaction, social preference, and social novelty preference. 2. Up-regulation of c-Fos expression in the mPFC by social stimulation was attenuated in IQSEC2 KO mice. 3. AMPAR, NMDAR, and GABAR-mediated synaptic transmissions were decreased in the pyramidal neurons in layer 5 of the mPFC in IQSEC2 KO mice. 4. These synaptic phenotypes were attributable to the postsynaptic deletion of IQSEC2 from the result of cell type specific IQSEC2 KD using in utero electroporation. 5. Re-expression of IQSEC2 isoform 1 in the mPFC rescued the electrophysiological and behavioral phenotypes in IQSEC2 KO mice.

Since our IQSEC2 KO mice harbor a stop codon in exon 3, the first common exon between isoform 1 and 2, both isoforms were completely removed. This is a similar condition to the mouse line generated by Jackson et al. [18]. Their heterozygote female KO mice exhibited hyperactivity, increased anxiety, and impaired spatial learning and memory. But they have not studied behaviors in hemizygote KO mice. Another KO mouse line, generated by Sah et al., has a stop codon in exon 7 [19]. This is also a part of common exons between isoform 1 and 2. They demonstrated hemizygote KO showed hyperactivity and elevated anxiety. Hyperactivity is commonly observed between three independent KO lines. But we failed to detect a change in anxiety levels in the elevated plus-maze. They have not analyzed center time in the open field test, another parameter for anxiety, but their sample trace did not show clear reduction in the traveling center time. Elevated anxiety level may be a mild phenotype, so it may be influenced by the background strain of mice or environmental condition of the behavioral experiment. Our IQSEC2 KO mice showed an increase in novel object preference (Figure 1F). Novel object preference is often considered to reflect hippocampus-dependent learning and memory ability. Our result is inconsistent with previous reports from heterozygote IQSEC2 KO female mice and IQSEC2 A350V mutant mice that show impaired learning and memory. Considering the high prevalence of ID in human patients with the IQSEC2 mutation, enhanced novel object preference in our IQSEC2 KO mice is unlikely to indicate improvement in learning and memory ability. It may be more plausible that it is due to the excessive interest in the inanimate, one of the autistic traits. Social behavior has not been studied in those two lines. Thus, this is the first demonstration that IQSEC2 deletion causes clear social deficits. The social deficit is frequently observed in human patients with IQSEC2 mutations as a core symptom of autism [8,36]. Given these behavioral phenotypes, IQSEC2 KO mice can be among a few animal models for non-syndromic X-linked autism [37].

c-Fos has been used to identify responsible brain regions upon behavioral tasks in animal models. The mPFC has been shown to regulate social behavior [38]. Our experiment reproduced the up-regulation of c-Fos expression in the mPFC after social behavioral task along with several other suggested regions. IQSEC2 KO selectively attenuated the c-Fos up-regulation in the mPFC. This suggests that IQSEC2 is required for the neural activity that may regulate social behavior in the mPFC.

In synapse physiology, we found both excitatory and inhibitory synaptic transmissions were impaired in the IQSEC2 KO and KD neurons. Both AMPAR and NMDAR-mediated synaptic transmissions were decreased in the excitatory synapses in IQSEC2 KO/KD neurons. This is consistent with the previous results using siRNA in organotypic hippocampal cultures [17] and acute slices [16]. IQSEC2 has been shown to regulate AMPAR in two different pathways. One is activity-dependent removal of GluA1-containing AMPAR in response to the calcium signal through NMDAR [14]. Another one is the activity-independent pathway that inserts GluA2-containing AMPAR in the postsynaptic membrane to maintain basal synaptic activity [17]. The synaptic phenotype observed in this study is likely to be due to the deficiency of the activity-independent pathway. Presynaptic phenotypes, such as impaired release probability in the excitatory synapse, were observed in IQSEC2 KO mice. Given that the same presynaptic phenotypes were observed in the pos-synapse-specific KD experiment using in utero electroporation, this is likely due to the transsynaptic effect of the IQSEC2 deficiency in the post-synapse. Retrograde signals from post- to pre-synapse have been suggested to maintain presynaptic functions. For instance, PSD-95 has been shown to regulate presynaptic release probability through the interaction with the postsynaptic cell adhesion molecule Neuroligin-1 [39]. Since IQSEC2 directly binds to PSD-95 [1], IQSEC2-PSD-95-Neuroligin-1 protein interaction may regulate presynaptic release probability. Endocannabinoid is also known to act as a retrograde signal that affects presynaptic release probability. β-Neurexins, presynaptic ligands for Neuroligins, have been shown to transsynaptically regulate postsynaptic endocannabinoid synthesis [40]. It is intriguing to investigate whether IQSEC2 is involved in these molecular mechanisms. In the KD experiment using in utero electroporation, we introduced shRNA for IQSEC2 in pyramidal neurons in layer 2/3 of the somatosensory cortex, not in layer 5 of the mPFC, for technical reason. Apart from the change in the amplitude of mEPSCs found in KD neurons, most of the electrophysiological phenotypes are identical between IQSEC2 KD and KO neurons. This may indicate that these synaptic effects of IQSEC2 may be common in pyramidal neurons in the different neural circuits in the brain.

GABAR-mediated synaptic transmission was also decreased in IQSEC2 KO/KD neurons. Considering the excitatory post-synapse specific localization of IQSEC2 protein, this result is enigmatic. One explanation may be due to the developmental effect of GluN2B. GluN2B has been shown to introduce GABAergic input during synapse development [29]. Given the notion that IQSEC2 is predominantly associated with GluN2B-containing NMDAR [16], GluN2B is likely to be down-regulated in IQSEC2 KO/KD neurons during synapse development. Unlike excitatory synapse, the release probability of inhibitory synapse was unaltered in IQSEC2 KO neurons. This also supports the idea that the number of functional GABAergic input may be decreased in IQSEC2 KO/KD neurons. Another simple explanation is it may be due to a compensatory homeostatic effect caused by strong excitatory phenotype of IQSEC2 KO and KD [41,42,43].

We found the re-expression of full-length IQSEC2 isoform 1 in the mPFC is sufficient to rescue the social behavior. We employed EFs promoter for driving the IQSEC2 gene in the AAV vector. EFs is a ubiquitous promoter that drives gene expression in all cell types within the AAV infected area. We focused on the pyramidal neurons in layer 5 of the mPFC for analyzing synapse physiology. Sah et al. identified that the glutamatergic input onto GABAergic neuron was selectively increased in the dissociated hippocampal culture prepared from their IQSEC2 KO mice [19]. To narrow down neuron types responsible for the social deficits within the mPFC, further studies using cell type specific promoters to introduce the IQSEC2 gene will be required.

## 5. Conclusions

We generated IQSEC2 KO mice and studied their behavior and electrophysiological properties focusing on the mPFC. The IQSEC2 KO mice exhibited overgrooming and social deficits reminiscent of the symptoms of autism. AMPAR, NMDAR, and GABAR-mediated synaptic transmissions were impaired in pyramidal neurons in layer 5 of the mPFC in IQSEC2 KO mice. Re-expression of IQSEC2 in the mPFC rescued both synaptic and social behavioral phenotypes, suggesting that an impairment in the neural function in the mPFC may be responsible for social deficits in IQSEC2 KO mice.

## Figures and Tables

**Figure 1 cells-10-02724-f001:**
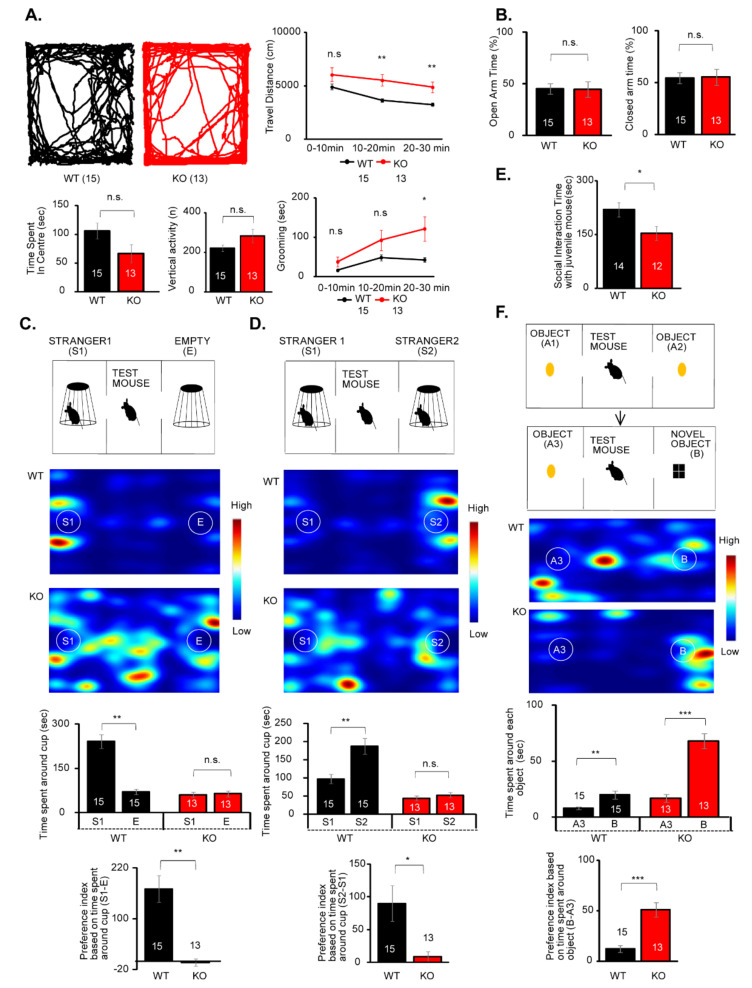
IQSEC2 KO mice display hyperactivity, overgrooming, impaired social and social novelty preference, and enhanced novel object preference. (**A**) Trajectories of movement and summary graphs for travel distance, time spent in center, vertical activity, and grooming in the open field test. Travel distance and grooming time are increased in IQSEC2 KO mice. (**B**) Summary graphs for elevated plus maze test. No changes were detected in time spent in open and closed arms between wild-type and IQSEC2 KO mice. (**C**) Experimental diagram (top), heatmap images (middle), and summary graphs (bottom) for social preference test using three-chamber apparatus. Preference for a stranger mouse (S1) to an empty cage (**E**) is diminished in IQSEC2 KO mice. (**D**) Experimental diagram (top), heatmap images (middle), and summary graphs (bottom) for social novelty preference test using three-chamber apparatus. Preference for a new stranger mouse (S2) to a familiar mouse (S1) is diminished in IQSEC2 KO mice. (**E**) Summary graph for interaction time with a juvenile mouse. Interaction time with a free-moving juvenile mouse is decreased in IQSEC2 KO mice. (**F**) Experimental diagram (top), heatmap images (middle), and summary graph (bottom) for novel object preference test. Preference for a novel object is increased in IQSEC2 KO mice. Data are means ± SEM (numbers of animals examined are shown in graphs). Statistical analyses were performed by Student’s *t-*test (* *p* < 0.05; ** *p* < 0.01; *** *p* < 0.001; n.s. = not significant). Data values used in graphs are shown in Appendix A.

**Figure 2 cells-10-02724-f002:**
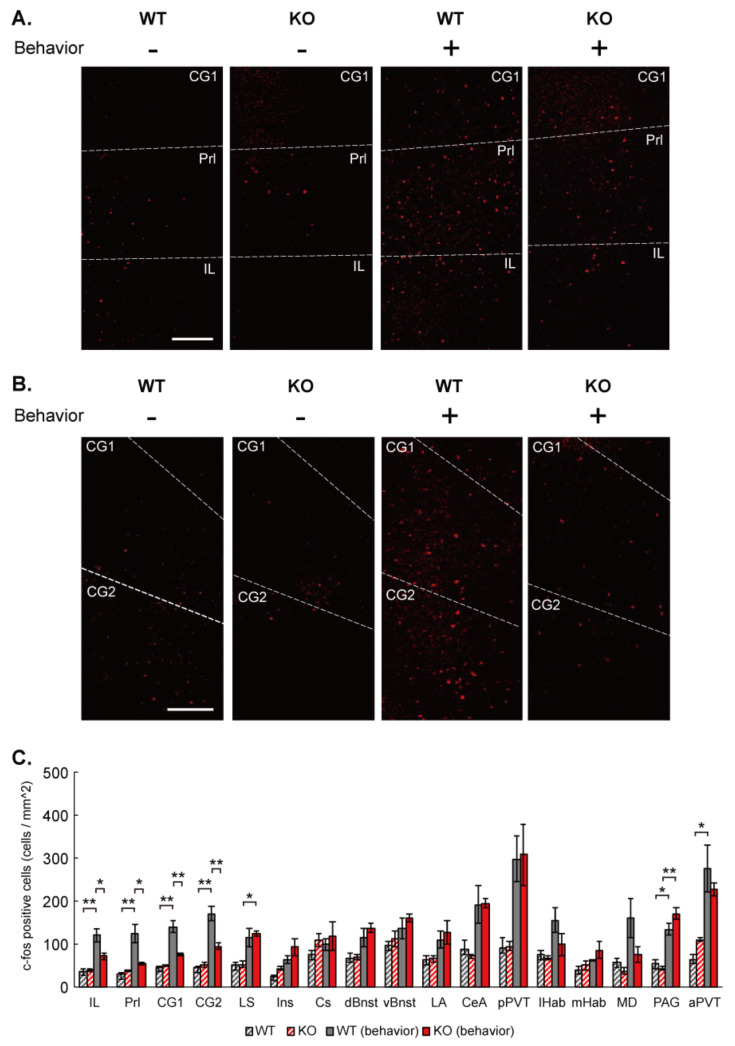
Elevated c-Fos expression in the mPFC by social stimulation is diminished in IQSEC2 KO mice. (**A**,**B**) Representative images for c-Fos immunostaining in the brain slices containing mPFC Figure 2. KO (KO) mice with and without imposing social behavior. Scale bar = 0.2 mm. (**C**) Summary graph for c-Fos expression in wild-type (WT) and IQSEC2 KO (KO) mice with and without imposing social behavior in different brain regions. The number of c-Fos positive neuron/mm^2^ is shown in the bar graph. c-Fos expression is up-regulated in IL, Prl, CG1, CG2, LS, PAG, aPVT in wild-type mice after social behavior. The up-regulation is diminished in IL, Prl, CG1, CG2. Abbreviations; IL: infralimbic cortex, Prl: prelimbic cortex, CG1: cingulate area 1, CG2: cingulate area 2, LS: lateral septum, Ins: insula, Cs: cingulate sulcus, dBnst: dorsal bed nucleus of the stria terminalis, vBnst: ventral bed nucleus of the stria terminalis, LA: lateral amygdala, CeA: central amygdala, pPVT: posterior paraventricular thalamus, lHaB: lateral habenula, mHaB: medial habenula, MD: medial dorsal thalamus, PAG: periaqueductal gray, aPVT: anterior paraventricular thalamus. Data in the graph are means ± SEM. Statistical analyses were performed by One-way ANOVA followed by Tukey’s post-hoc test (* *p* < 0.05; ** *p* < 0.01. Data values used in graphs are shown in Appendix A.

**Figure 3 cells-10-02724-f003:**
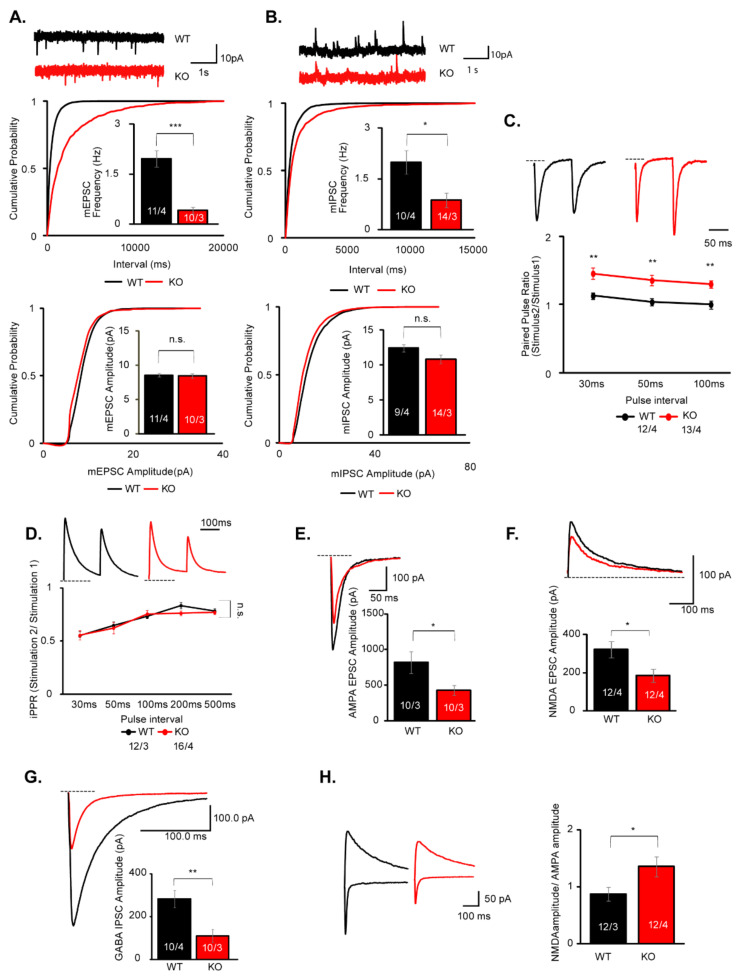
AMPAR, NMDAR, and GABAR-mediated synaptic transmissions are decreased in pyramidal neurons in layer 5 of the mPFC in P14-P19 IQSEC2 KO mice. (**A**) Sample traces and summary graphs for mEPSCs in pyramidal neurons in layer 5 of the mPFC in wild-type (WT) and IQSEC2 KO (KO) mice. The frequency, but not amplitude, of mEPSCs is decreased in IQSEC2 KO mice. (**B**) Sample traces and summary graphs for mIPSCs in pyramidal neurons in layer 5 of the mPFC in wild-type (WT) and IQSEC2 KO (KO) mice. The frequency, but not amplitude, of mIPSCs is decreased in IQSC2 KO mice. (**C**) Sample traces and summary graph for the paired-pulse ratio of evoked EPSC in wild-type (WT) and IQSEC2 KO (KO) mice. The paired-pulse ratio is increased in IQSEC2 KO mice in 30, 50, and 100 ms stimulation intervals, suggesting that the release probability is decreased in the IQSEC2 KO neurons. (**D**) Sample traces and summary graph for the paired-pulse ratio of evoked IPSC in wild-type (WT) and IQSEC2 KO (KO) mice. The paired-pulse ratio is unchanged in IQSEC2 KO mice. (**E**) Sample traces and summary graph for evoked AMPA EPSC in wild-type (WT) and IQSEC2 KO (KO) mice. The amplitude of evoked AMPA EPSC is decreased in IQSEC2 KO mice. (**F**) Sample traces and summary graph for evoked NMDA EPSC in wild-type (WT) and IQSEC2 KO (KO) mice. The amplitude of evoked NMDA EPSC is decreased in IQSEC2 KO mice. (**G**) Sample traces and summary graph for evoked GABA IPSC in wild-type (WT) and IQSEC2 KO (KO) mice. The amplitude of evoked GABA IPSC is decreased in IQSEC2 KO mice. (**H**) Sample traces and summary graph for the NMDA/AMPA ratio in wild-type (WT) and IQSEC2 KO (KO) mice. The NMDA/AMPA ratio is increased in IQSEC2 KO mice. Data are means ± SEM (numbers of neurons/independent animals examined are shown in graphs). Statistical analyses were performed by Student’s *t*-test (* *p* < 0.05; ** *p* < 0.01; *** *p* < 0.001; n.s. = not significant). Data values used in graphs are shown in Appendix A.

**Figure 4 cells-10-02724-f004:**
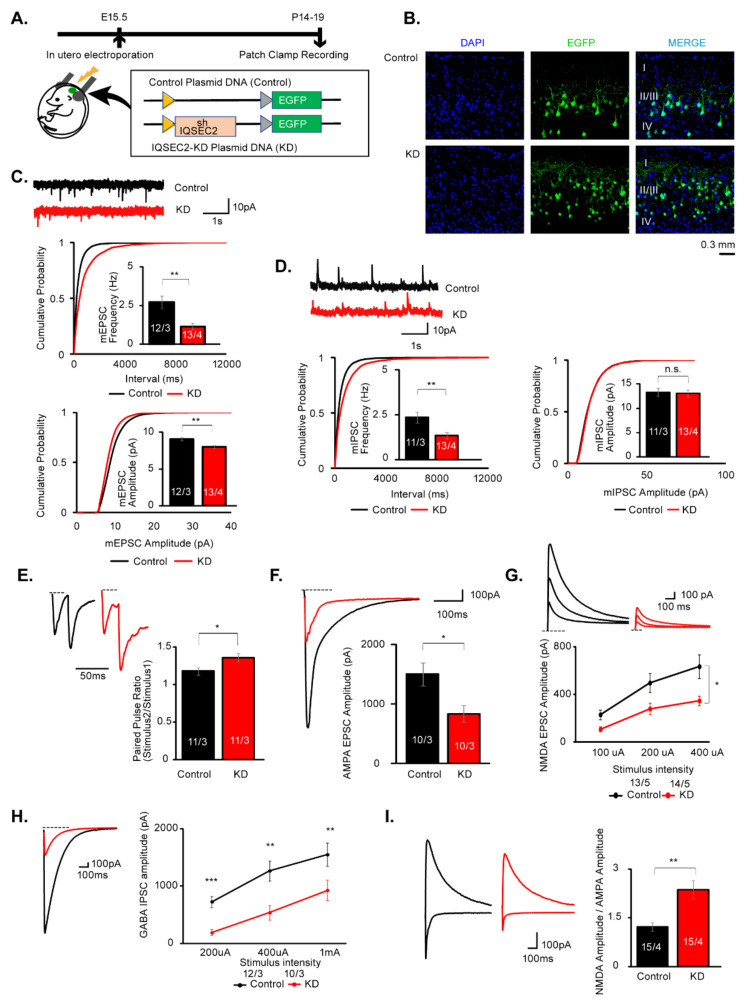
AMPAR, NMDAR, and GABAR-mediated synaptic transmissions are decreased in IQSEC2 KD pyramidal neurons in layer 2/3 of the somatosensory cortex. (**A**) Schematic diagrams for the experimental design and plasmid DNAs for Control and IQSEC2-KD used for the IQSEC2 KD experiment. (**B**) Confocal microscopic images for control and IQSEC2-KD plasmid transfected mouse brain sections. EGFP signals are observed sparsely in pyramidal neurons in layer 2/3 somatosensory cortex. Scale bar = 0.3 mm. (**C**) Sample traces and summary graphs for mEPSCs in control (Control) and IQSEC2 KD (KD) neurons. Both frequency and amplitude of mEPSCs are decreased in IQSEC2 KD neurons. (**D**) Sample traces and summary graphs for mIPSCs in control (Control) and IQSEC2 KD (KD) neurons. The frequency, but not amplitude, of mEPSCs is decreased in IQSEC2 KD neurons. (**E**) Sample traces and summary graph for the paired-pulse ratio of evoked EPSCs in control (Control) and IQSEC2 KD (KD) neurons. The paired-pulse ratio is increased in IQSEC2 KD neurons, suggesting release probability is decreased by suppression of IQSEC2 gene expression. (**F**) Sample traces and summary graphs for evoked AMPA EPSC in control (Control) and IQSEC2 KD (KD) neurons. The amplitude of evoked AMPA EPSC is decreased in IQSEC2 KD neurons. (**G**) Sample traces and summary graph for evoked NMDA EPSC in control (Control) and IQSEC2 KD (KD) neurons. The amplitude of evoked NMDA EPSC is decreased in IQSEC2 KD neurons. (**H**) Sample traces and summary graph for evoked GABA IPSC in control (Control) and IQSEC2 KD (KD) neurons. The amplitude of evoked GABA IPSC is decreased in IQSEC2 KD neurons. (**I**) Sample traces and summary graph for the NMDA/AMPA ratio in wild-type (WT) and IQSEC2 KD (KD) neurons. The NMDA/AMPA ratio is significantly increased in IQSEC2 KD neurons. Data are means ± SEM (numbers of neurons/independent animals examined are shown in graphs). Statistical analyses were performed by Student’s *t*-test (* *p* < 0.05; ** *p* < 0.01; *** *p* < 0.001; n.s. = not significant). Data values used in graphs are shown in Appendix A.

**Figure 5 cells-10-02724-f005:**
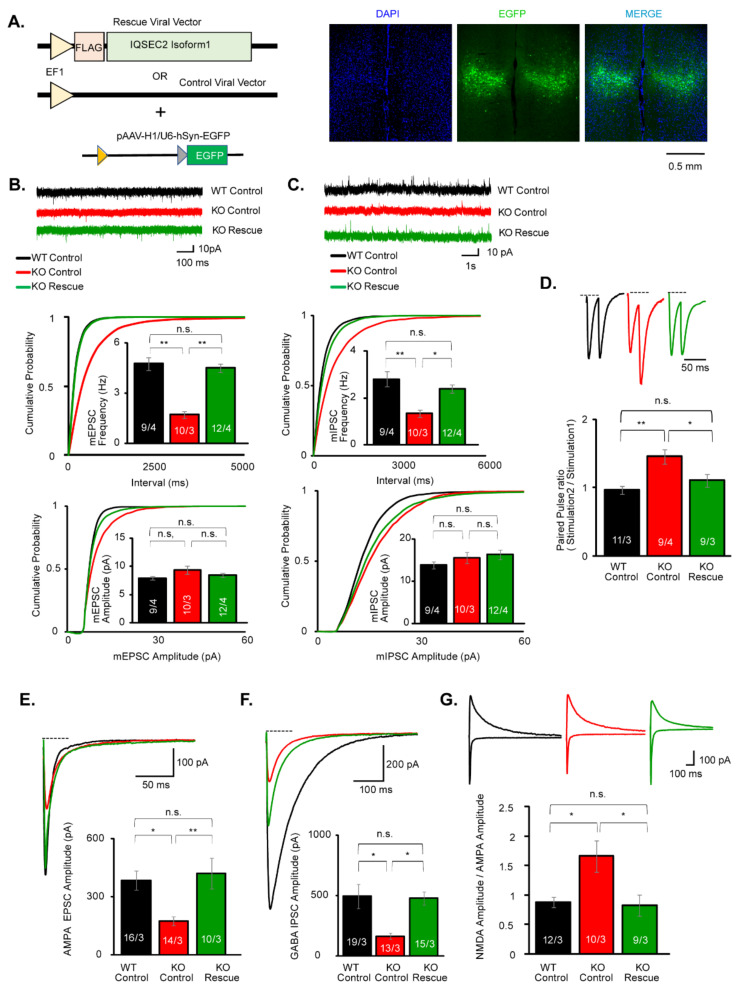
Re-expression of IQSEC2 isoform 1 rescues the impaired synaptic transmission in the pyramidal neurons in layer 5 of the mPFC in IQSEC2 KO mice. (**A**) Schematic diagram for AAV vectors used for IQSEC2 rescue, control, and EGFP marker (left). Representative confocal microscopic images of AAV-injected brain slice containing both hemispheres of mPFC. Scale bar = 0.5 mm. (**B**) Sample traces and summary graphs for mEPSCs in wild-type infected with control AAV (WT Control), IQSEC2 KO infected with control AAV (KO Control), and IQSEC2 KO infected with IQSEC2 AAV (KO Rescue) pyramidal neurons in layer 5 of the mPFC. The decreased frequency of mEPSCs in IQSEC2 KO neurons is recovered in KO Rescue neurons. (**C**) Sample traces and summary graphs for mIPSCs in wild-type infected with control AAV (WT Control), IQSEC2 KO infected with control AAV (KO Control), and IQSEC2 KO infected with IQSEC2 AAV (KO Rescue) pyramidal neurons in layer 5 of the mPFC. The decreased frequency of mIPSCs in IQSEC2 KO neurons is rescued in KO Rescue neurons. (**D**) Sample traces and summary graph for the paired-pulse ratio of evoked AMPA EPSCs in wild-type infected with control AAV (WT Control), IQSEC2 KO infected with control AAV (KO Control), and IQSEC2 KO infected with IQSEC2 AAV (KO Rescue) pyramidal neurons in layer 5 of the mPFC. The increased paired-pulse ratio is restored to the wild-type level in the KO Rescue neurons. (**E**) Sample traces and summary graph for evoked AMPA EPSC in wild-type infected with control AAV (WT Control), IQSEC2 KO infected with control AAV (KO Control), and IQSEC2 KO infected with IQSEC2 AAV (KO Rescue) neurons. The decreased amplitude of AMPA EPSC is recovered in the KO Rescue neurons. (**F**) Sample traces and summary graph for evoked GABA IPSC in wild-type infected with control AAV (WT Control), IQSEC2 KO infected with control AAV (KO Control), and IQSEC2 KO infected with IQSEC2 AAV (KO Rescue) neurons. The decreased amplitude of GABA IPSC is recovered in the KO rescue neurons. (**G**) Sample traces and summary graph for the NMDA/AMPA ratio in wild-type infected with control AAV (WT Control), IQSEC2 KO infected with control AAV (KO Control), and IQSEC2 KO infected with IQSEC2 AAV (KO Rescue) neurons. The increased NMDA/AMPA ratio is restored in KO Rescue neurons. Data are means ± SEM (numbers of neurons/independent animals examined are shown in graphs). Statistical analyses were performed by One-way ANOVA followed by Bonferroni post-hoc test (* *p* < 0.05; ** *p* < 0.01; n.s. = not significant. Data values used in graphs are shown in Appendix A.

**Figure 6 cells-10-02724-f006:**
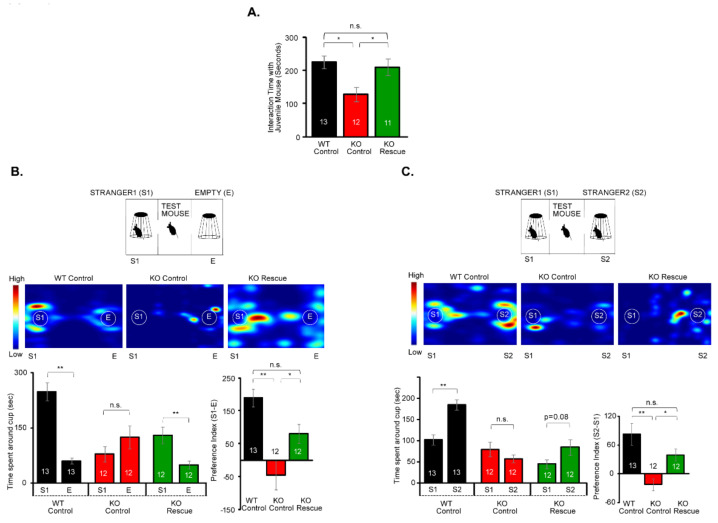
Re-expression of IQSEC2 in the mPFC rescues social behaviors. (**A**) Summary graph for interaction with a free-moving juvenile mouse in wild-type and IQSEC2 KO mice infected with control AAV (WT Control and KO Control), and IQSEC2 KO mice infected with IQSEC2 AAV (KO Rescue). Interaction with a juvenile mouse is restored in the KO rescue mice. (**B**) Experimental diagram (top), heatmap images (middle), and summary graphs (bottom) for social preference test in wild-type infected with control AAV (WT Control), IQSEC2 KO infected with control AAV (KO Control), and IQSEC2 KO mice infected with IQSEC2 AAV (KO Rescue). Preference for stranger mouse (S1) is restored in KO Rescue mice. (**C**) Experimental diagram (top), heatmap images (middle), and summary graphs (bottom) for social novelty preference test in wild-type infected with control AAV (WT Control), IQSEC2 KO infected with control AAV (KO Control), and IQSEC2 KO mice infected with IQSEC2 AAV (KO Rescue). Preference for a new stranger mouse (S2) is restored in KO Rescue mice. Data are means ± SEM (numbers of animals examined are shown in graphs). Statistical analyses were performed by One-way ANOVA followed by Tukey’s post-hoc test in A, Student’s *t*-test in B and C (time spent around cup), One-way ANOVA followed by Tukey’s post-hoc test in B and C (preference index). **p* < 0.05; ***p* < 0.01; n.s. = not significant. Data values used in graphs are shown in Appendix A.

## Data Availability

Most of the data generated and analyzed during this study are included in this published article. All datasets used and/or analyzed during the current study are available from the corresponding author on reasonable request.

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
