# Peer review of "IQSEC2 Deficiency Results in Abnormal Social Behaviors Relevant to Autism by Affecting Functions of Neural Circuits in the Medial Prefrontal Cortex"

_cells, 2021, doi:10.3390/cells10102724_

Round 1
Reviewer 1 Report
Submitted manuscript presents data from meticulously planned and very well executed experiments aimed to assess the impact of IQSEC2 knockout on the behavior and synapse function in mice. The results are very interesting and suitable for publication.
Author Response
We deeply thank the reviewer for the positive evaluation.
Reviewer 2 Report
Overall, this is an interesting and well driving research article in which Mehta et al., generated a IQSEC2 KO mice (model of some X-linked neurodevelopmental disorders, such as autism) and demonstrated that this mice model exhibited overgrooming and social deficits, as well as, impaired excitatory and inhibitory synaptic transmissions in pyramidal neurons in layer 5 of the mPFC. Besides, they demonstrated that the reexpression of IQSEC2 in the mPFC rescued both synaptic and social behavioral phenotypes, leading to the conclusion that an impairment in the neural function in the mPFC may be responsible for social deficits in these animals.
However, I have several comments:
- Please check all the abbreviations. In general, terms should not be abbreviated unless they are used repeatedly, and the abbreviation is helpful to the reader. For instance, electroconvulsive threshold test (ECT), pentylenetetrazole (PTZ) only appear one time and the abbreviations are not necessary.
Furthermore, you should review that you use the word or phrase in full when you refer the term for the first time followed by the abbreviation in parentheses and thereafter use the abbreviation only. For instance, medial prefrontal cortex appears, initially, in material and methods in the text and you should abbreviate in here. Another example is the terms knockout and the abbreviation KO. In the introduction you directly put KO but in material and method you state knockout (KO). Please check also PPR, WT, NMDR, AMPAR….
- How do you identified pyramidal neurons of layer V? Did you use soma morphology in IR-DIC before the patch recording? Did you performed a post-patch identification using neuronal morphology in dye-filled cells (more accurate measure)? Or did you use another criterion? You must address this issue in material and methods.
- As I understand, to know if the decreased release probability was due to a presynaptic function of IQSEC2 or postsynaptic IQSEC2 you knocked down IQSEC2 expression specifically in postsynaptic neurons by in utero electroporation, but you introduce the shRNAs into pyramidal neurons in layer 2/3 of the somatosensory cortex and not in pyramidal neurons from layer V in prefrontal cortex. You state that you made that because of technical accessibility. Could this fact underlie the differences found between KO and KD? Did you expect the same results in somatosensory cortex than in prefrontal cortex? Maybe you could include a little paragraph, or a sentence addressed this issue in discussion.
Author Response
We deeply thank the reviewer for the positive evaluation. We changed our manuscript by following the reviewers suggestions as follows.
- Please check all the abbreviations. In general, terms should not be abbreviated unless they are used repeatedly, and the abbreviation is helpful to the reader. For instance, electroconvulsive threshold test (ECT), pentylenetetrazole (PTZ) only appear one time and the abbreviations are not necessary.
Thank you for the critical point. We removed (ETC) and (PTZ) from the text.
Furthermore, you should review that you use the word or phrase in full when you refer the term for the first time followed by the abbreviation in parentheses and thereafter use the abbreviation only. For instance, medial prefrontal cortex appears, initially, in material and methods in the text and you should abbreviate in here. Another example is the terms knockout and the abbreviation KO. In the introduction you directly put KO but in material and method you state knockout (KO). Please check also PPR, WT, NMDR, AMPAR….
We spelled out AMPAR, NMDAR, and GABAR in the first appearance in the Abstract and Introduction section and used abbreviations after that. We also corrected for mPFC, KO, KD, ACSF, PPR, ePPR, and iPPR in the same way. We decided not to abbreviate the term 'wild-type' and removed (WT) from the firstly appeared sentence in the Materials and Methods section.
- How do you identified pyramidal neurons of layer V? Did you use soma morphology in IR-DIC before the patch recording? Did you performed a post-patch identification using neuronal morphology in dye-filled cells (more accurate measure)? Or did you use another criterion? You must address this issue in material and methods.
We identified pyramidal neurons of layer 5 morphologically in IR-DIC before the patch-clamp recording. We added the sentence “Acute brain slices were transferred to a recording chamber continuously superfused with oxygenated ACSF (1.5 ml/min) maintained at 30.5 ° C. For patch-clamp electrophysiological recording, pyramidal neurons in layer 5 of the mPFC were identified morphologically using an infrared-differential interference contrast microscope (BX50WI; Olympus, Tokyo, Japan) with a ×40 water immersion objective and a charge-coupled device camera (C3077-79; Hamamatsu Photonics, Hamamatsu, Japan). “ in the Materials and Methods section.
- As I understand, to know if the decreased release probability was due to a presynaptic function of IQSEC2 or postsynaptic IQSEC2 you knocked down IQSEC2 expression specifically in postsynaptic neurons by in utero electroporation, but you introduce the shRNAs into pyramidal neurons in layer 2/3 of the somatosensory cortex and not in pyramidal neurons from layer V in prefrontal cortex. You state that you made that because of technical accessibility. Could this fact underlie the differences found between KO and KD? Did you expect the same results in somatosensory cortex than in prefrontal cortex? Maybe you could include a little paragraph, or a sentence addressed this issue in discussion.
Thank you for the critical suggestion. We added the sentences “In the KD experiment using in utero electroporation, we introduced shRNA for IQSEC2 in pyramidal neurons in layer 2/3 of the somatosensory cortex, not in layer 5 of the mPFC, for technical reason. Apart from the change in the amplitude of mEPSCs found in KD neurons, most of the electrophysiological phenotypes are identical between IQSEC2 KD and KO neurons. This may indicate that these synaptic effects of IQSEC2 may be common in pyramidal neurons in the different neural circuits in the brain.” at the end of the fourth paragraph in the Discussion section. We also modified the sentence “Considering the result of the postsynapse-specific knockdown experiment, this is likely due to the transsynaptic effect of the IQSEC2 deficiency in the postsynapse.” to “Given that the same presynaptic phenotypes were observed in the postsynapse-specific KD experiment using in utero electroporation, this is likely due to the transsynaptic effect of the IQSEC2 deficiency in the postsynapse. “ in the middle of the fourth paragraph to emphasize the observation of presynaptic phenotypes in IQSEC2 KD in pyramidal neurons in layer 2/3 of the somatosensory cortex.